# High Uric Acid Levels in Acute Myocardial Infarction Provide Better Long-Term Prognosis Predictive Power When Combined with Traditional Risk Factors

**DOI:** 10.3390/jcm11195531

**Published:** 2022-09-21

**Authors:** Soohyun Kim, Byung-Hee Hwang, Kwan Yong Lee, Chan Jun Kim, Eun-Ho Choo, Sungmin Lim, Jin-Jin Kim, Ik Jun Choi, Mahn-Won Park, Gyu Chul Oh, Ki Dong Yoo, Wook Sung Chung, Youngkeun Ahn, Myung Ho Jeong, Kiyuk Chang

**Affiliations:** 1Cardiovascular Center and Cardiology Division, Seoul St. Mary’s Hospital, The Catholic University of Korea, Seoul 06591, Korea; 2Cardiovascular Center and Cardiology Division, Uijeongbu St. Mary’s Hospital, The Catholic University of Korea, Uijeonbu 11765, Korea; 3Cardiovascular Center and Cardiology Division, Incheon St. Mary’s Hospital, The Catholic University of Korea, Incheon 21431, Korea; 4Cardiovascular Center and Cardiology Division, Daejeon St. Mary’s Hospital, The Catholic University of Korea, Daejeon 34943, Korea; 5Cardiovascular Center and Cardiology Division, St. Vincent’s Hospital, The Catholic University of Korea, Suwon 16247, Korea; 6Department of Cardiology, Cardiovascular Center, Chonnam National University Hospital, Gwangju 61469, Korea

**Keywords:** uric acid, acute myocardial infarction, heart failure, prognosis, risk factor, clinicaltrials.gov NCT 02806102

## Abstract

The current study aimed to investigate the association between serum UA levels and the mortality rate of AMI patients. We analyzed 5888 patients with successfully revascularized AMI (mean age: 64.0 ± 12.7 years). The subjects were divided into the high UA group (uric acid >6.5 mg/dL for males, >5.8 mg/dL for females) or the normal UA group based on initial serum UA level measured at admission. The primary outcome was all-cause mortality. A total of 4141 (70.3%) and 1747 (29.7%) patients were classified into the normal UA group and high UA groups, respectively. Over a median follow-up of 5.02 (3.07, 7.55) years, 929 (21.5%) and 532 (34.1%) patients died in each group. Cox regression analysis identified high UA levels as an independent predictor of all-cause mortality (unadjusted hazard ratio (HR) 1.69 [95% CI 1.52–1.88]; *p* < 0.001, adjusted HR 1.18 [95% CI: 1.05–1.32]; *p* = 0.005). The results were consistent after propensity-score matching and inverse probability weighting to adjust for baseline differences. The predictive accuracies of conventional clinical factor discrimination and reclassification were significantly improved upon the addition of hyperuricemia (C-index 0.788 [95% CI 0.775–0.801]; *p* = 0.005, IDI 0.004 [95% CI 0.002–0.006]; *p* < 0.001, NRI 0.263 [95% CI 0.208–0.318]; *p* < 0.001).

## 1. Introduction

In modern times, faster and more successful reperfusion treatment is currently being performed by developing drug-eluting stent technology, antiplatelet agents, and emergency medical systems. Nevertheless, survivors of acute myocardial infarction (AMI) have high morbidity and mortality. Therefore, identifying poor prognostic factors related to AMI patients will be helpful as patients can be observed more carefully, and optimal prophylactic management can be performed. We noted in previous studies that increased serum uric acid (UA) levels might be related to high levels of cardiovascular mortality [1]. UA is the end-product of purine metabolism metabolized by xanthine oxidase [2,3] and reflects xanthine oxidase activity [4,5]. It is not clear how UA correlates with coronary heart disease, but some studies demonstrate that circulating uric acid as a result of coronary reperfusion impairments is likely to represent a new biomarker in determining the prognosis of coronary heart disease [6]. Reperfusion impairment of the myocardium due to coronary artery disease activates xanthine oxidase circulating in the endothelium, increasing serum uric acid levels and resulting in endothelium dysfunction [4,5,6,7,8].

A previous systematic review and meta-analysis demonstrated that hyperuricemia may marginally increase the risk of coronary heart disease events, independent of traditional coronary heart disease risk factors [9,10,11]. Over the past few years, several studies have explored the value of serum UA in hospitalization to predict outcomes in patients with acute coronary syndrome [12,13]. Nevertheless, not all epidemiological studies support this hypothesis [14]. In some studies, after additional adjustment for cardiovascular disease risk factors, uric acid levels were no longer associated with coronary heart disease [15,16]. Several authors have suggested that hyperuricemia is a risk indicator rather than an independent risk factor [17]. Elevated serum UA levels are difficult to determine if they are a result or a cause of hypertension, diabetes mellitus, dyslipidemia, metabolic syndrome, chronic kidney disease, and various risk factors [18,19,20,21,22]. This is because serum uric acid is also associated with diverse etiological risk factors for cardiovascular disease that can confound observed associations [23]. In addition, it is still debatable whether hyperuricemia is an independent predictor of patients with percutaneous coronary intervention in the AMI cohort [15]. Some studies showed different results in the association between hyperuricemia and other comorbid conditions [16,24,25,26]. Moreover, data on whether the predictive value can be increased in addition to the traditional cardiovascular risk factors are lacking. New prediction models adding UA in recent studies were significantly improved at calibration and discrimination, but the increase in AUC was weak and not statistically significant [6,27]. This study aimed to investigate the long-term prognostic role of uric acid in patients with AMI who underwent successful revascularization therapy.

## 2. Materials and Methods

### 2.1. Study Protocols and Population Selection

The Convergent REgistry of cAtholic and chonnAm University for Acute MI (COREA-AMI) registry was designed to evaluate real-world, long-term clinical outcomes in all consecutive patients with AMI at nine major cardiac centers in Korea. All hospitals perform a large number of percutaneous coronary interventions (PCIs) in AMI patients and are located throughout the country. The COREA-AMI I registry included AMI patients undergoing PCI from January 2004 to December 2009, and the COREA-AMI II registry extended the follow-up period of COREA-AMI I patients and enrolled additional AMI patients from January 2010 to August 2014. The clinical, angiographic, and follow-up data of all AMI patients were consecutively registered in the electronic, web-based case report system. The COREA-AMI study was conducted in accordance with the Declaration of Helsinki. This observational study was approved by the institutional review board of our institution (IRB No. XC15RSMI0089K) and performed in accordance with the Strengthening the Reporting of Observational Studies in Epidemiology guidelines [28]. The COREA-AMI registry is registered on ClinicalTrials.gov (NCT02806102). In total, 10,719 patients with AMI who underwent PCI with drug-eluting stents (DESs) were enrolled in the registry, and patients who did not undergo serum uric acid testing at admission (*n* = 4377) and those without echo data before revascularization (*n* = 454) were excluded from the analysis. Thus, 5888 patients were selected for this analysis. The study flowchart is depicted in Figure 1. Patients were divided into a high UA group (uric acid >6.5 mg/dL for males, >5.8 mg/dL for females) or a normal UA group based on the initial serum UA level measured at the time of AMI.

### 2.2. Treatment and Data Collection

All patients underwent PCI within 48 h after admission. Coronary angiography and primary PCI were performed according to the current standard guidelines. Significant coronary artery disease was defined by angiographic stenosis ≥70% in the epicardial coronary arteries and ≥50% in the left main coronary artery. A loading dose of the antiplatelet agent (aspirin, 300 mg; clopidogrel, 300 mg or 600 mg; cilostazol, 200 mg; ticagrelor, 180 mg; or prasugrel, 60 mg) was prescribed for all patients before or during PCI. Patients with DESs were prescribed P2Y12 inhibitors (clopidogrel, 75 mg once daily; ticagrelor, 90 mg twice daily; or prasugrel, 10 mg once daily) and/or aspirin at 100 mg daily. The duration of dual antiplatelet agent administration was determined by a physician in accordance with the final diagnosis at baseline and the revascularization procedure complexity. Optimal pharmacological therapy, including statins, beta-blockers, angiotensin-converting enzyme (ACE) inhibitors, or angiotensin II receptor blockers (ARBs) was recommended according to the guidelines. Doses were titrated, and medications were changed during follow-up, if needed due to each patient’s condition. Predilation, direct stenting, postadjunct balloon inflation, and glycoprotein IIb/IIIa receptor blocker administration were performed at the discretion of individual physicians.

All data were collected in a web-based system after removing personal information. Patient follow-up data, including survival data and clinical event data, were collected through 31 March 2019, via hospital chart reviews and telephone interviews of the patients conducted by trained reviewers who were blinded to the study results. Independent reviewers and interventional cardiologists assessed the angiographic and procedural data, and independent research personnel collected baseline clinical, laboratory, and medication data. Each patient was followed up at 1, 6, and 12 months and then annually thereafter. All adverse clinical events of interest were confirmed centrally by the committee of the Cardiovascular Center of Seoul St. Mary’s Hospital (Seoul, Korea). Validation of mortality was performed on the basis of disqualification from the National Health Insurance Service, which is the single government-managed insurance provider, covering almost all of the nation’s population. The final dataset was handled by independent statisticians at the clinical research coordinating center and sealed with a code by the clinical research associate.

### 2.3. Study Endpoints and Definition

The primary endpoint of this analysis was all-cause mortality. The secondary ischemic outcomes included cardiac death, recurrent MI, readmission for heart failure, readmission for unstable angina, any revascularization, and ischemic stroke. Cardiac death was defined as death resulting from AMI, sudden cardiac death, heart failure, stroke, or other vascular causes. Recurrent MI was defined as the presence of recurrent symptoms and new ECG changes that were compatible with MI or cardiac markers that were expressed at least twofold above the normal limit. Clinically driven revascularization that occurred after discharge from the index hospitalization was coded as a revascularization event, according to the Academic Research Consortium definitions. Ischemic stroke was defined as an episode of neurologic dysfunction related to the brain, spinal cord, or retinal vascular injury because of infarction. The Global Registry of Acute Coronary Events (GRACE) risk score was used to assess the ischemic risk [29].

### 2.4. Statistical Analysis

Categorical variables were presented as numbers and relative frequencies (percentages) and were compared using the Chi-squared test or Fisher’s exact test. Continuous variables were expressed as mean ± standard deviation, and were compared using the independent sample *t*-test. The optimal cutoff value of UA in our study was determined using ROC curve analysis with Youden index. The thresholds for hyperuricemia in our study were >6.5 mg/dL (386.6 µmol/L) for males and >5.8 mg/dL (345.0 µmol/L) for females. 

The cumulative event rates of each group were calculated using a Kaplan–Meier estimator and compared using the log-rank statistic. Because differences in the baseline characteristics could significantly affect outcomes, sensitivity analyses were performed to adjust for confounders as much as possible. First, to identify independent predictors of all-cause mortality, we used a multivariable Cox proportional hazards model. The adjusted variables for the multivariate model were selected if they were significantly different between the two groups (*p* < 0.05 in the univariable analysis) for the baseline characteristics. The adjusted variables for the multivariate Cox proportional hazards regression analysis were age ≥65 years, sex, body mass index (BMI) >25, diabetes mellitus, hypertension, dyslipidemia, history of stroke, current smoker, atrial fibrillation, estimated glomerular filtration rate (eGFR) <30, chronic lung disease, left ventricular ejection fraction (LVEF) ≤35%, ST-segment elevation MI (STEMI), clopidogrel, prasugrel, potent P2Y12 inhibitor, angiotensin-converting enzyme inhibitor (ACEi) or angiotensin II receptor blocker (ARB), oral anticoagulation, hemoglobin, creatinine, triglyceride, high-density lipoprotein (HDL), low-density lipoprotein (LDL), and multivessel disease. Second, Cox proportional hazard regression in a propensity score (PS)-matched cohort and inverse probability weighted (IPW) Cox proportional hazard regression was performed. Propensity scores were obtained from logistic regression with a significant difference between the two groups. For the propensity-score matching, a 1:1 matching process without replacements was performed by a greedy nearest-neighbor matching algorithm with a caliper width of 0.2 standard deviations, yielding 1538 patients in the high UA group matched with 1538 control subjects in the normal UA group. The PS-matching analysis included the following covariates: age ≥65 years, sex, BMI >25, diabetes mellitus, hypertension, dyslipidemia, history of stroke, current smoker, atrial fibrillation, eGFR <30, chronic lung disease, LVEF ≤35%, STEMI, clopidogrel, prasugrel, potent P2Y12 inhibitor, ACEi or ARB, oral anticoagulation, hemoglobin, creatinine, triglyceride, HDL, LDL, multivessel disease. For the IPW adjustment, the inverse of the propensity score was adjusted using the proportional hazard regression model (Appendix A). Balance between the 2 groups after propensity-score matching or IPW adjustment was assessed by calculating percent standardized mean differences. The percent standardized mean differences after propensity-score matching were within ±10% across all matched covariates, demonstrating successful achievement of balance between the comparative groups.

Various prediction models were constructed to assess the incremental prognostic value of high UA and applied to all-cause mortality data: (1) Model 1, A, A′, A″: conventional clinical risk factors; (2) Model 2, B, B′, B″: each of conventional clinical risk factors models + high UA. Conventional clinical risk factors were based on the published risk factors for mortality in AMI, such as GRACE score, old age (age ≥65 years), sex, obesity (body mass index >25), hypertension, diabetes mellitus, dyslipidemia, prior stroke, smoking, atrial fibrillation, chronic lung disease, chronic kidney disease (estimated glomerular filtration rate <30), and impaired left ventricular function (LV ejection fraction ≤35%). The discriminative ability of the models was assessed using Harrell’s C-index, which is analogous to the area under the receiver operator characteristic curve and was applied to all-cause mortality data. The receiver operating characteristic (ROC) curves in logistic regression were used. Reclassification performance was compared using the relative integrated discrimination improvement (IDI) and continuous net reclassification index (NRI). We used model Akaike information criterion (AIC) values as the parameters of model fit. Larger relative IDI values and smaller AIC values indicate greater improvements in model discrimination. Improvements in subject risk reclassification were further assessed using continuous NRI and were applied to the all-cause mortality data. Each measure was analyzed using R version 4.1.2 (R Foundation for Statistical Computing, Vienna, Austria). The survival package in R was used for survival analysis. The pROC package was used to interpret a ROC curve in logistic regression in R. Statistical significance was indicated by a two-tailed *p* < 0.05.

## 3. Results

### 3.1. Baseline Characteristics

The baseline clinical, medications at discharge, laboratory, and angiographic characteristics are listed in Table 1 and Table 2. The mean age of all included patients was 64.0 ± 12.7 years, and 70.3% were male. Overall, 37.9% had diabetes mellitus, 71.0% had hypertension, 7.0% had dyslipidemia with eGFR <30 mL/min/1.73 m^2^ and 35.1% had obesity with BMI over 25 kg/m^2^. Regarding angiographic lesion and procedural profiles, only 3.6% of patients presented with cardiogenic shock. The mean LVEF was 52.7 ± 11.6%. Overall, 52.6% of patients were diagnosed with ST-segment elevation MI. Of the 5888 included patients, 4326 patients were classified into the normal UA group, and 1562 were classified into the high UA group. The patients in the high UA group were more likely to be older, female, obese, current nonsmokers, with diabetes mellitus, hypertension, a history of stroke, atrial fibrillation, chronic kidney disease, chronic lung disease, impaired LV systolic function, and STEMI and use prasugrel, beta-blockers, ACEi or ARB less often and clopidogrel and oral anticoagulant more often (Table 1). Regarding laboratory data, the patients in the high UA group had higher levels of uric acid, creatinine, and triglycerides but lower levels of hemoglobin and high-density lipoprotein cholesterol than those in the normal UA group. Regarding angiographic findings, the high UA group had more multivessel disease, bifurcation PCI with two stents, and use of secondary-generation drug-eluting stents (Table 2). 

### 3.2. Primary Outcomes

Over a median follow-up of 5.02 (3.07, 7.55) years, 929 (21.5%) and 532 (34.1%) patients died in the normal and high UA groups, respectively. Kaplan–Meier analysis and univariate Cox proportional hazard model showed that the all-cause mortality rate was significantly higher in the high UA group than in the intermediate score group (21.5% vs. 34.1%; unadjusted HR 1.69 [95% CI 1.52–1.88]; *p* < 0.001) (Figure 2). The difference in the mortality results mainly involved cardiac death (15.9% vs. 26.7%, *p* < 0.001). The readmission rate for heart failure was also significantly higher in the high UA group (6.3%) than in the normal UA group (3.7%) (*p* < 0.001). However, no significant differences in the event rates of recurrent myocardial infarction, readmission for unstable angina, stent thrombosis, or ischemic stroke were noted (*p* = 0.068, 0.063, 0.094, and 0.121, respectively). Concordant results were shown in the sensitivity analyses, including multivariate adjustment, P-S matching, and IPW adjustment (Table 3). The risk of revascularization was significantly higher in the high UA group with univariate Cox regression; however, the risk changed, showing a nonsignificant difference after adjusting for the influence of multivariate variables. Multivariate Cox regression demonstrated that age ≥65, LVEF ≤35%, atrial fibrillation, eGFR <30 mL/min/1.73 m^2^, history of stroke, diabetes mellitus, creatinine, anemia, body mass index, HDL cholesterol, and triglycerides were significant predictors of all-cause mortality after adjustment (*p* < 0.05, for each) (Appendix A). Interestingly, the use of ACEi or ARB and potent P2Y12 inhibitor was independently associated with a decreased risk of all-cause death (adjusted HR: 0.55; [95% CI: 0.49–0.62]; *p* < 0.001, HR 0.50 [95% CI 0.30–085]; *p* = 0.01, respectively).

### 3.3. Risk Prediction, Discrimination, and Reclassification

Receiver operating characteristic analysis was performed to evaluate the availability of the serum uric acid level to predict mortality in patients with AMI. The sensitivity and specificity of the serum uric acid levels at a cutoff of >6.5 mg/dL (386.6 µmol/L) were 34.79 and 76.97% for males, >5.8 mg/dL (345.0 µmol/L), and 38.81 and 76.06% for females, respectively. Figure 3 and Table 4 shows the incremental value of a high serum uric acid level over conventional risk factors using conventional parameter of model fit (model AIC value) and model global performance (changes in C-statistic, IDI, and continuous NRI). The addition of a high serum uric acid level to the GRACE score, a well-validated conventional cardiovascular risk score in AMI, significantly increased the discriminant ability to predict mortality compared with the GRACE score alone (C-index: 0.741, 95% CI: 0.727–0.756, *p* = 0.008) (Figure 3). Regarding predictions of all-cause death, the high uric acid addition model (Model 2) showed a significantly higher reclassification ability compared with the GRACE score model (Model 1) (NRI: 0.008, 95% CI: 0.005–0.01, *p* < 0.001; IDI: 0.263 95% CI: 0.208–0.318, *p* < 0.001). Table 4 depicts the effect of adding a high serum uric acid level to well-known clinical risk factors on prediction accuracy and risk reclassification ability. For predicting all-cause death, the addition of high UA levels to all of the various combination models of clinical risk factors (Model B, B′, B″) showed a significantly higher discriminant and reclassification ability against the risk factors alone models (Model A, A′, A″) (*p* < 0.01 for all). 

The addition of high uric acid to GRACE score showed significantly higher discriminant and reclassification abilities for all-cause mortality than the GRACE score alone.

## 4. Discussion

In the present study, we compared the five-year clinical outcomes of patients with AMI between the high UA group and the normal UA group using data from a large multicenter observational study. The main findings were as follows. First, the prevalence of hyperuricemia was 26.5% in AMI participants in our study. Despite guideline-directed drug treatment and the use of more than a majority of second-generation DESs, the five-year mortality rate of hyperuricemia patients was greater than 30%. Second, the optimal cutoff value of the serum uric acid level for mortality was >6.5 mg/dL (386.6 µmol/L) for males and >5.8 mg/dL (345.0 µmol/L) for females. Third, the high UA group showed a significantly higher mortality risk than the normal UA group (Figure 2), which was consistently observed after thorough sensitivity analyses for adjustment of baseline differences (Table 3). Fourth, the performance of a risk prediction model based on C-statistics was significantly improved upon the addition of hyperuricemia to models composed of traditional clinical risk factors (Figure 3 and Table 4). Both NRI and IDI analyses further supported this finding.

### 4.1. Pathophysiology: Uric Acid and the Development of Cardiovascular Diseases

Uric acid (UA) is the end-product of purine metabolism metabolized by xanthine oxidase [2,3], and UA levels reflect xanthine oxidase activity [4,5]. We noted in previous studies that increased serum uric acid levels might be related to high levels of cardiovascular mortality. Although the mechanism by which UA leads to cardiovascular (CV) events has not been well demonstrated, it is suggested that oxidative stress leads to endothelial dysfunction during the conversion of hypoxanthine into xanthine, generates superoxide anions, and increases oxidative stress, which is a known atherosclerotic risk factor. Elevated UA levels are associated with lipid-rich plaques, reduced coronary flow reserve, and impaired coronary microvascular function, factors known to be associated with future adverse outcomes [4,5,7,8]. Hence, the association between serum uric acid (UA) and cardiovascular diseases, including hypertension [2], metabolic syndrome [20], coronary artery disease [30], cerebrovascular disease [19], vascular dementia [31], preeclampsia [32], and kidney disease [21,22], was revealed. UA has been presumed to be linked with cardiovascular disease. Since the relationship between UA and cardiovascular events was demonstrated in the Framingham study, which included 5127 patients, and found an increased risk of myocardial infarction (MI) in patients with hyperuricemia [1], various studies have reported an association between serum uric acid and cardiovascular outcomes. Our results are consistent with previous studies that showed a significant relationship between UA levels and poor clinical outcomes, revealing several potential confounding factors [9,13]. The Rotterdam study, which enrolled 4385 patients and investigated the incidence of MI prospectively for an average of 8.4 years, reported that high serum UA levels were associated with the risk of MI (HR 1.87, 194 cases, 1.12 to 3.13, 95% CI) and stroke (381 cases, 1.57, 1.11 to 2.22, 95% CI) after adjusting for age and sex [33]. Kojima et al. evaluated 1124 patients hospitalized for acute MI and reported higher mortality in hyperuricemia (HR 3.7) [34]. A meta-analysis study (237,433 patients) showed a marginal association between hyperuricemia and coronary heart disease mortality (RR: 1.209 [95% CI: 1.003–1.457]; *p* = 0.047), but an increased risk for coronary heart disease incidence (RR: 1.206 [95% CI: 1.066–1.364]; *p* = 0.003) [9]. 

### 4.2. Limited Evidence Regarding Hyperuricemia as a Cardiovascular Risk Factor

However, UA as an independent direct cardiovascular risk factor remains controversial. To date, limited studies with small populations have been conducted, and the results were not concordant in different subjects. Some studies have reported a correlation between UA and AMI [12,13,35,36,37]. However, it is still debatable whether hyperuricemia is an independent predictor of patients with percutaneous coronary intervention in the AMI cohort [15]. Some authors mentioned that serum UA may not play a causal role in cardiovascular disease and is simply an indicator of the presence of risk factors such as hypertension, diabetes mellitus, metabolic syndrome, or chronic kidney disease [15]. This debate prompted the conduct of different studies in populations with comorbidities, with conflicting results in specific subsets [16,25,38]. A subgroup analysis of our study confirmed that the independent association between hyperuricemia and total mortality was still significant in participants with a GFR ≥ 30 but not in participants with an eGFR <30 mL/min/1.73 m^2^ (Appendix A). However, this study focuses on the intensity of hyperuricemia as a prognostic indicator. In the future, prospective studies of patients divided into specific groups based on sex, diabetes mellitus, hypertension, and eGFR may be needed.

### 4.3. The Optimal Cutoff Value of Uric Acid Level in AMI Population

Hyperuricemia is commonly defined by an increase in the circulating blood as a concentration of uric acid in serum above the threshold level of 7 mg/dL (416.4 μmol/L) in men and 6 mg/dL (356.9 μmol/L) in women, and these cutoffs have been widely applied in various studies [12,37,39]. However, the necessity of identifying and validating an optimal cutoff for cardiovascular risk is emerging because the conventional cutoff was adopted from studies with gouty conditions. A previous study showed that UA >6.5 mg/dL was observed in 21.5% of patients and was independently linked with in-hospital mortality in STEMI patients undergoing PCI [25]. The optimal cutoff value in our study was determined using ROC curve analysis with the Youden index, and the association with long-term mortality was evaluated in a large multicenter AMI cohort. The thresholds for hyperuricemia in our study were >6.5 mg/dL (386.6 µmol/L) for males and >5.8 mg/dL (345.0 µmol/L) for females. High uric acid defined by our cutoff was independently associated with an increased risk of all-cause death in both males (adjusted HR: 1.61; [95% CI: 1.41–1.85]; *p* < 0.001) and females (adjusted HR: 1.79; [95% CI: 1.51–2.11]; *p* < 0.001). The cutoff obtained through statistical analysis in our study was similar to the optimal UA cutoff proposed in the previous study results [40]. According to a previous subgroup study, uric acid had a higher CHD mortality rate in women (RR: 1.47, 95% CI: 1.21–1.73) compared with men (RR: 1.47, 95% CI: 1.00–1.19) [41]. In a previous systematic review and meta-analysis, subgroup analysis showed no significant association between high uric acid and cardiac death rate in men, but an increased risk for cardiac death in hyperuricemic women (RR: 1.446, 95% CI: 1.323–1.581) [9]. 

### 4.4. Predictive Value of the Hyperuricemia in Patients with AMI

In our study, Table 4 and Figure 3 show the incremental value of high uric acid levels over conventional risk factors using conventional parameters of model global performance. The addition of a high uric acid level to any of the conventional risk factor combination models significantly improved the predictive accuracy for discrimination (changes in the C-statistic, IDI). Moreover, the addition model was significantly better at predicting the probability of reclassification (changes in continuous NRI). In other previous studies, they tried to test predictive power in a similar manner as described in our study [6,27]. However, in one previous study using the GRACE score, the reclassification and discrimination of new models with added UA were improved, but the AUC increased only slightly without statistical significance [6]. In our study, the addition of UA was significant in that it showed an improvement in predictive power when applied to a model including all 12 well-known clinical risk factors as well as GRACE scores (Table 4 and Figure 3). The reason may be that the population in our study included AMI patients with a higher risk than that noted for ACS patients. In fact, the annual mortality rate was higher in our data by a few percent points. The clinical risk factors used in our study included old age, sex, obesity, HBP, DM, dyslipidemia, stroke, smoking, AF, CLD, CKD, and low LVEF. Because well-known risk factors already show good risk prediction, it is not easy for a new factor in addition to the existing risk prediction model to significantly improve prediction accuracy [42]. In addition, identifying independent risk factors in a cohort of high-risk patients represents a more clinically meaningful discovery. In particular, our study population showed high-risk characteristics, including a mean GRACE score of 137.4 ± 44.1 and the presence of several comorbidities, such as arterial hypertension (71.0%), diabetes mellitus (37.9%), CKD with eGFR <30 mL/min/1.73 m^2^ (7.0%), and LVEF ≤35% (8.0%). In addition, greater than half (54.3%) of the patients had multivessel disease, and the mean total stent length was quite long (37.6 ± 25.0 mm). 

### 4.5. Limitations

First, this study used retrospective observational cohort data. Our findings need validation from prospectively designed research or randomized controlled trials with large populations in the future. Second, we used uric acid levels obtained from blood tests conducted on the day of hospitalization for acute myocardial infarction. Blood test levels at this time may be affected by stress induced by myocardial infarction, and the conclusions of our study cannot be extended to the results of uric acid levels measured at different times. In addition, we excluded 40.8% of patients without uric acid level results from the analysis. This process can induce selection bias. Third, although we have presented results by various statistical methodologies that adjust differences in confounding factors described in the baseline table, differences in results may occur due to differences in unmeasured confounding factors. However, to minimize this difference, we have tried to include as many combinations of factors as possible, including medication, laboratory, and procedural findings that can affect mortality. Fourth, the cohort data used in our analysis did not include the presence of gout history before hospitalization and the use of drugs that reduce UA.

## 5. Conclusions

Our study was conducted on AMI patients who received successful reperfusion treatment. Greater than half (59.4%) of patients received newer generation DES treatment and optimal drug treatment recommended by the current guideline. Nevertheless, 34.1% of people with hyperuricemia died over five years, and a significantly higher mortality rate was observed compared with the normal UA population. When using the cutoff of UA (>6.5 mg/dL (386.6 µmol/L) for males and >5.8 mg/dL (345.0 µmol/L) for females) we found that high UA was an independent risk factor for a high cardiovascular mortality rate. In addition, high UA has resulted in a significant improvement in risk prediction ability in addition to previously well-known cardiovascular disease risk factors.

## Figures and Tables

**Figure 1 jcm-11-05531-f001:**
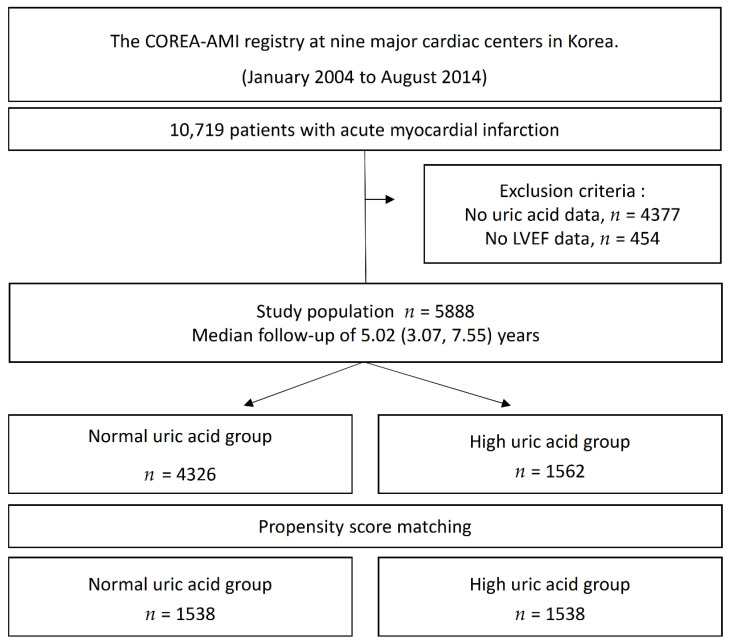
Study Flow.

**Figure 2 jcm-11-05531-f002:**
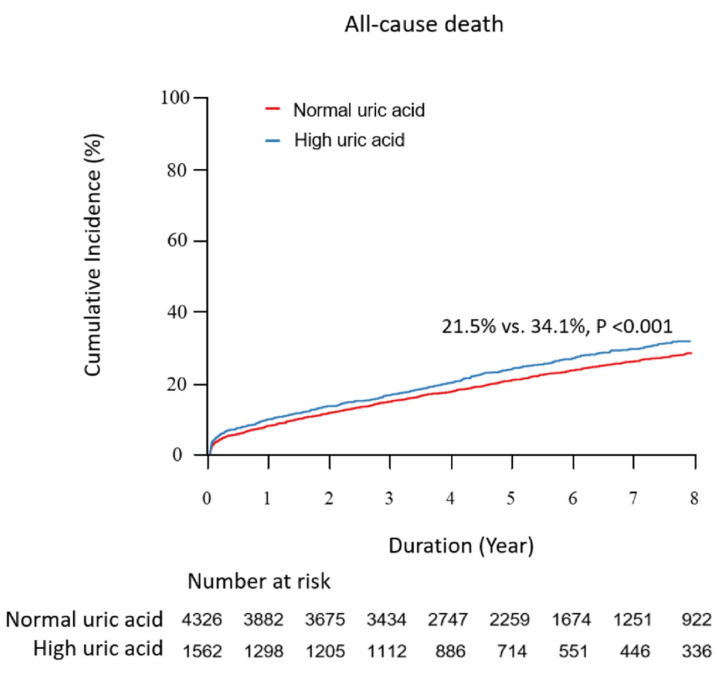
Kaplan–Meier Curve with Cumulative Hazards of All-cause Death Compared According to the Uric Acid Level.

**Figure 3 jcm-11-05531-f003:**
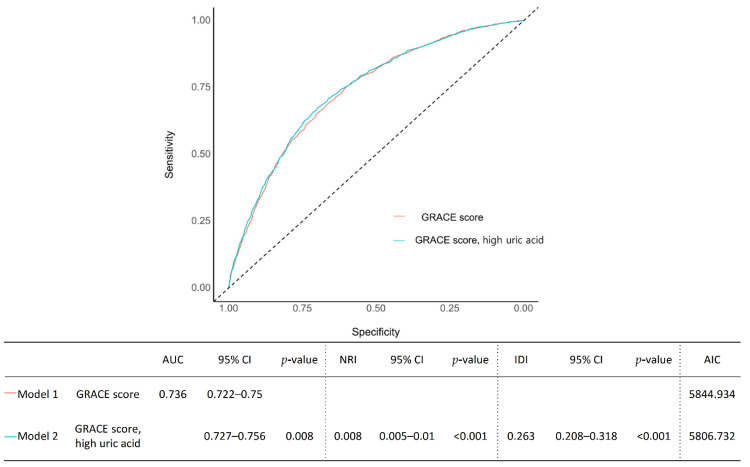
Prognostic Impact of Serum Uric Acid Level in Patients with Acute Myocardial Infarction.

**Table 1 jcm-11-05531-t001:** Baseline Clinical and Medication Characteristics.

		Before PS-Matching				After PS-Matching			
	Total (*n* = 5888)	Normal Uric Acid (*n* = 4326)	High Uric Acid (*n* = 1562)	*p*-Value	Absolute SMD	Normal Uric Acid (*n* = 1538)	High Uric Acid (*n* = 1538)	*p*-Value	Absolute SMD
Clinical characteristics									
Age, years	64.0 ± 12.7	63.6 ± 12.4	65.4 ± 13.3	<0.001	0.141	65.0 ± 12.4	65.3 ± 13.3	0.487	0.025
Age ≥65 year	3010 (51.1)	2119 (49.0)	891 (57.0)	<0.001	0.162	844 (54.9)	876 (57.0)	0.26	0.042
Male	4141 (70.3)	3085 (71.3)	1056 (67.6)	0.007	0.081	1049 (68.2)	1039 (67.6)	0.728	0.014
BMI, kg/m2	24.1 ± 3.2	24.0 ± 3.2	24.3 ± 3.4	0.022	0.069	24.3 ± 3.3	24.3 ± 3.4	0.713	0.013
BMI >25 kg/m2	2069 (35.1)	1485 (34.3)	584 (37.4)	0.032	0.064	588 (38.2)	576 (37.5)	0.683	0.016
DM	2232 (37.9)	1584 (36.6)	648 (41.5)	0.001	0.1	630 (41.0)	634 (41.2)	0.912	0.005
Hypertension	4182 (71.0)	3008 (69.5)	1174 (75.2)	<0.001	0.126	1151 (74.8)	1151 (74.8)	1	<0.001
History of dyslipidemia	1371 (23.3)	1013 (23.4)	358 (22.9)	0.716	0.012	362 (23.5)	351 (22.8)	0.669	0.017
History of Stroke	472 (8.0)	326 (7.5)	146 (9.3)	0.027	0.065	145 (9.4)	143 (9.3)	0.951	0.004
Current smoker	2280 (38.7)	1724 (39.9)	556 (35.6)	0.003	0.088	569 (37.0)	551 (35.8)	0.524	0.024
Previous MI	259 (4.4)	187 (4.3)	72 (4.6)	0.688	0.014	68 (4.4)	69 (4.5)	1	0.003
Previous PCI	456 (7.7)	327 (7.6)	129 (8.3)	0.406	0.026	129 (8.4)	127 (8.3)	0.948	0.005
Previous CABG	32 (0.5)	22 (0.5)	10 (0.6)	0.685	0.017	10 (0.7)	10 (0.7)	1	<0.001
Atrial fibrillation on baseline ECG	224 (3.8)	144 (3.3)	80 (5.1)	0.002	0.089	85 (5.5)	78 (5.1)	0.629	0.02
eGFR <30	412 (7.0)	183 (4.2)	229 (14.7)	<0.001	0.362	181 (11.8)	214 (13.9)	0.085	0.064
Chronic liver disease	59 (1.0)	43 (1.0)	16 (1.0)	1	0.003	18 (1.2)	16 (1.0)	0.863	0.012
Chronic lung disease	153 (2.6)	101 (2.3)	52 (3.3)	0.043	0.06	62 (4.0)	52 (3.4)	0.39	0.034
Cancer	198 (3.4)	142 (3.3)	56 (3.6)	0.626	0.017	53 (3.4)	56 (3.6)	0.845	0.011
LVEF	52.7 ± 11.6	53.4 ± 11.1	50.5 ± 12.6	<0.001	0.248	50.6 ± 12.6	50.7 ± 12.5	0.89	0.005
LVEF ≤35%	505 (8.6)	288 (6.7)	217 (13.9)	<0.001	0.24	227 (14.8)	207 (13.5)	0.325	0.037
ST-segment elevation MI	3095 (52.6)	2313 (53.5)	782 (50.1)	0.023	0.068	801 (52.1)	775 (50.4)	0.367	0.034
Medication at discharge									
Aspirin	5793 (98.4)	4258 (98.4)	1535 (98.3)	0.761	0.012	1504 (97.8)	1511 (98.2)	0.438	0.033
Clopidogrel	5151 (87.5)	3761 (86.9)	1390 (89.0)	0.04	0.063	1361 (88.5)	1369 (89.0)	0.69	0.016
Ticagrelor	331 (5.6)	240 (5.5)	91 (5.8)	0.73	0.012	94 (6.1)	90 (5.9)	0.82	0.011
Prasugrel	371 (6.3)	301 (7.0)	70 (4.5)	0.001	0.107	66 (4.3)	69 (4.5)	0.86	0.01
Potent P2Y12 inhibitor	702 (11.9)	541 (12.5)	161 (10.3)	0.024	0.069	160 (10.4)	159 (10.3)	1	0.002
Beta-blocker	5131 (87.1)	3824 (88.4)	1307 (83.7)	<0.001	0.137	1283 (83.4)	1291 (83.9)	0.733	0.014
ACEi or ARB	4443 (75.5)	3333 (77.0)	1110 (71.1)	<0.001	0.137	1096 (71.3)	1093 (71.1)	0.937	0.004
Oral anticoagulant	154 (2.6)	97 (2.2)	57 (3.6)	0.004	0.083	57 (3.7)	55 (3.6)	0.923	0.007
Statin	5638 (95.8)	4154 (96.0)	1484 (95.0)	0.102	0.049	1457 (94.7)	1463 (95.1)	0.681	0.018

Data are presented as the *n* (%) for categorical variables and as the mean ± standard deviation for continuous variables. eGFR = 141 ∗ min(Scr/κ,1)α ∗ max(Scr/κ, 1) − 1.209 ∗ 0.993 Age ∗ 1.018 (if female) ∗ 1.159 (if black). Scr is serum creatinine (mg/dL), κ is 0.7 for females and 0.9 for males, α is −0.329 for females and −0.411 for males, min indicates the minimum of Scr/κ or 1, and max indicates the maximum of Scr/κ or 1. PS indicates propensity score; IPW, inverse probability weighted; SMD, standardized mean differences; BMI, body mass index; DM, diabetes mellitus; MI, myocardial infarction; CABG, coronary artery bypass graft; ECG, electrocardiography; eGFR, estimated glomerular filtration rate; LVEF, left ventricle ejection fraction; ACEi, angiotensin-converting enzyme inhibitors; ARB, angiotensin II receptor blockers.

**Table 2 jcm-11-05531-t002:** Baseline Laboratory and Angiographic Characteristics.

		Before PS-Matching				After PS-Matching			
	Total (*n* = 5888)	Normal Uric Acid (*n* = 4326)	High Uric Acid (*n* = 1562)	*p*-Value	Absolute SMD	Normal Uric Acid (*n* = 1538)	High Uric Acid (*n* = 1538)	*p*-Value	Absolute SMD
Laboratory findings									
Uric acid, mg/dL	5.6 ± 4.0	4.3 ± 1.4	9.1 ± 6.1	<0.001	1.071	4.3 ± 1.4	9.1 ± 6.2	<0.001	1.065
Uric acid, µmol/L	(333.1 ± 237.9)	(255.8 ± 83.3)	(541.3 ± 362.8)			(255.8 ± 83.3)	(541.3 ± 368.8)		
CK-MB, peak, ng/mL	130.0 ± 753.8	124.4 ± 267.3	145.2 ± 1394.5	0.558	0.021	130.5 ± 378.1	146.6 ± 1405.2	0.665	0.016
Hemoglobin, mg/dL	13.4 ± 2.2	13.5 ± 2.1	13.1 ± 2.5	<0.001	0.154	13.2 ± 2.2	13.1 ± 2.5	0.284	0.039
Creatinine, mg/dL	1.2 ± 1.2	1.1 ± 1.0	1.5 ± 1.5	<0.001	0.344	1.4 ± 1.6	1.5 ± 1.4	0.081	0.063
high-sensitivity CRP, mg/dL	5.3 ± 18.5	5.1 ± 18.1	5.7 ± 19.6	0.286	0.032	6.6 ± 22.8	5.7 ± 19.7	0.236	0.043
Total cholesterol, mg/dL	176.8 ± 43.2	176.7 ± 42.6	177.1 ± 45.0	0.772	0.009	174.5 ± 44.6	177.5 ± 45.1	0.065	0.067
Triglyceride, mg/dL	122.2 ± 89.4	119.2 ± 84.4	130.3 ± 101.7	<0.001	0.119	132.4 ± 107.7	130.7 ± 102.2	0.644	0.017
High-density lipoprotein, mg/dL	40.6 ± 10.8	41.0 ± 10.9	39.4 ± 10.7	<0.001	0.149	39.1 ± 10.4	39.5 ± 10.7	0.359	0.033
Low-density lipoprotein, mg/dL	112.3 ± 37.1	112.5 ± 37.0	111.8 ± 37.6	0.52	0.019	110.1 ± 37.6	112.0 ± 37.7	0.167	0.05
Angiographic characteristics									
Multivessel disease	3198 (54.3)	2259 (52.2)	939 (60.1)	<0.001	0.16	881 (57.3)	921 (59.9)	0.153	0.053
Left main PCI	230 (3.9)	156 (3.6)	74 (4.7)	0.057	0.057	70 (4.6)	68 (4.4)	0.931	0.006
Left anterior descending PCI	3515 (59.7)	2595 (60.0)	920 (58.9)	0.471	0.022	934 (60.7)	902 (58.6)	0.255	0.042
Left circumflex PCI	1589 (27.0)	1188 (27.5)	401 (25.7)	0.183	0.041	438 (28.5)	396 (25.7)	0.096	0.061
Right coronary artery PCI	2364 (40.1)	1718 (39.7)	646 (41.4)	0.269	0.033	630 (41.0)	636 (41.4)	0.855	0.008
Total stent number	1.6 ± 0.9	1.6 ± 0.9	1.6 ± 0.9	0.103	0.048	1.6 ± 1.0	1.6 ± 0.9	0.72	0.013
Total stent length	37.6 ± 25.0	37.2 ± 24.8	38.6 ± 25.5	0.05	0.058	38.5 ± 25.8	38.5 ± 25.3	0.949	0.002
Bifurcation PCI with two stents	91 (1.5)	57 (1.3)	34 (2.2)	0.025	0.066	36 (2.3)	31 (2.0)	0.621	0.022
Long stenting >60 mm	283 (4.8)	209 (4.8)	74 (4.7)	0.937	0.004	85 (5.5)	73 (4.7)	0.369	0.035
Restenosis lesion	91 (1.5)	67 (1.5)	24 (1.5)	1	0.001	24 (1.6)	22 (1.4)	0.882	0.011
Ostial lesion	232 (3.9)	158 (3.7)	74 (4.7)	0.07	0.054	65 (4.2)	73 (4.7)	0.542	0.025
Second-generation DES	3499 (59.4)	2616 (60.5)	883 (56.5)	0.007	0.08	869 (56.5)	872 (56.7)	0.942	0.004

Data are presented as the *n* (%) for categorical variables and as the mean ± standard deviation for continuous variables. PS indicates propensity score; IPW, inverse probability weighted; SMD, standardized mean differences; CK-MB, creatine kinase muscle brain; CRP, C-reactive protein; PCI, percutaneous coronary intervention; DES, drug-eluting stent.

**Table 3 jcm-11-05531-t003:** Cumulative Ischemic Outcomes in AMI Patients According to Uric Acid Level.

	Normal Uric Acid (*n* = 4326)	High Uric Acid (*n* = 1562)	Unadjusted		Multivariable-Adjusted		Propensity-Score Matched		IPW-Adjusted	
			HR (95% CI)	*p*-Value ^†^	HR (95% CI)	*p*-Value	HR (95% CI)	*p*-Value ^†^	HR (95% CI)	*p*-Value ^†^
All cause of death	929 (21.5)	532 (34.1)	1.69 (1.52–1.88)	<0.001	1.18 (1.05–1.32)	0.005	1.19 (1.05,1.35)	0.008	1.18 (1.05,1.33)	0.005
Cardiac death	688 (15.9)	417 (26.7)	1.79 (1.58–2.02)	<0.001	1.23 (1.08–1.4)	0.002	1.23 (1.06,1.42)	0.005	1.2 (1.05,1.37)	0.009
Readmission for HF	158 (3.7)	98 (6.3)	1.86 (1.45–2.39)	<0.001	1.78 (1.31–2.41)	<0.001	1.14 (0.85,1.54)	0.368	1.28 (0.97,1.67)	0.076
Readmission for UA	470 (10.9)	132 (8.5)	0.83 (0.69–1.01)	0.063	0.79 (0.64–0.96)	0.018	0.8 (0.63,1)	0.051	0.82 (0.66,1)	0.051
Recurrent MI	231 (5.3)	96 (6.1)	1.25 (0.98–1.58)	0.068	1.02 (0.79–1.31)	0.892	1.01 (0.76,1.33)	0.968	1.06 (0.83,1.37)	0.629
Definite or probable ST	66 (1.5)	32 (2.0)	1.43 (0.94–2.19)	0.094	1.46 (0.94–2.26)	0.092	1.64 (0.94,2.88)	0.081	1.42 (0.91,2.2)	0.121
Revascularization	659 (15.2)	254 (16.3)	1.17 (1.01–1.35)	0.036	1.06 (0.91–1.23)	0.464	1.02 (0.85,1.21)	0.857	1.05 (0.9,1.22)	0.572
Ischemic stroke	124 (2.9)	54 (3.5)	1.29 (0.94–1.77)	0.121	1.13 (0.81–1.58)	0.47	1.15 (0.78,1.7)	0.484	1.08 (0.77,1.51)	0.655

Values are number of events (%) unless otherwise indicated. ^†^ *p* value from univariate Cox regression. The variables of multivariate analysis: Age ≥65, gender, BMI >25, diabetes mellitus, hypertension, dyslipidemia, history of stroke, current smoker, atrial fibrillation, eGFR <30, chronic lung disease, LVEF ≤35%, STEMI, clopidogrel, prasugrel, potent P2Y12 inhibitor, ACE inhibitor or ARB, oral anticoagulation, hemoglobin, creatinine, triglyceride, high-density lipoprotein, low-density lipoprotein, multivessel disease. HR indicates hazard ratio; CI, confidence interval; IPW, inverse probability weighted; HF, heart failure; UA, unstable angina; MI, myocardial infarction; ST, stent thrombosis.

**Table 4 jcm-11-05531-t004:** Effects of Variables on the Prediction Accuracy and Risk Reclassification of Each Model (Traditional Risk Factors Only vs. Traditional Risk Factors + High Uric Acid).

Model		C-Index	95% CI	*p*-Value	NRI	95% CI	*p*-Value for NRI	IDI	95% CI	*p*-Value for IDI	AIC
For predicting mortality											
Model A	Old age, gender, obesity, HBP, DM, dyslipidemia, stroke, smoker, AF, CLD	0.75	0.736–0.763								5703.343
Model B	Old age, gender, obesity, HBP, DM, dyslipidemia, stroke, smoker, AF, CLD, high UA	0.759	0.745–0.772	<0.001	0.012	0.008–0.015	<0.001	0.263	0.208–0.318	<0.001	5644.191
Model A′	Old age, gender, obesity, HBP, DM, dyslipidemia, stroke, smoker, AF, CLD, CKD	0.773	0.76–0.786								5495.279
Model B′	Old age, gender, obesity, HBP, DM, dyslipidemia, stroke, smoker, AF, CLD, CKD, high UA	0.779	0.766–0.792	<0.001	0.005	0.003–0.008	<0.001	0.263	0.208–0.318	<0.001	5468.476
Model A″	Old age, gender, obesity, HBP, DM, dyslipidemia, stroke, smoker, AF, CLD, CKD, low LVEF	0.785	0.772–0.797								5413.460
Model B″	Old age, gender, obesity, HBP, DM, dyslipidemia, stroke, smoker, AF, CLD, CKD, low LVEF, high UA	0.788	0.775–0.801	0.005	0.004	0.002–0.006	<0.001	0.263	0.208–0.318	<0.001	5395.117

Old age defined as age ≥65. Obesity defined as body mass index >25. CKD defined as estimated glomerular filtration rate <30. High UA defined as uric acid >6.5 mg/dL (386.6 µmol/L) for males and >5.8 mg/dL (345.0 µmol/L) for females. low LVEF defined as left ventricular ejection fraction ≤35%. NRI indicates net reclassification index; CI, confidence interval; IDI, integrated discrimination improvement; AIC, Akaike information criterion; HBP, high blood pressure; DM, diabetes mellitus; AF, atrial fibrillation; CLD, chronic lung disease; CKD, chronic kidney disease; UA, uric acid.

## Data Availability

The datasets used and/or analyzed during the current study are available from the corresponding author upon reasonable request.

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
