# Peer review of "High Uric Acid Levels in Acute Myocardial Infarction Provide Better Long-Term Prognosis Predictive Power When Combined with Traditional Risk Factors"

_jcm, 2022, doi:10.3390/jcm11195531_

Round 1
Reviewer 1 Report
In the submitted paper the authors present an important analysis of the effect of serum UA levels on mortality of patients with AIM. The study population is robust with sufficient follow-up time, the methodology is adequate.
I recommend to indicate the cutoff value for UA (discrimination between high UA and normal group) both in mg/dl and μmol/l
It would be useful to analyse whether patients were taking medication to reduce UA (e.g. allopurinol) and had a history of gout.
I recommend the paper for publication as a useful contribution to the study of biomarkers in AIM.
Reviewer 2 Report
Soohyun Kim and coll. analyzed the prognostic significance of the hyperuricemia (HUA) in a cohort of patients with AMI from COREA-AMI studies and they found that HUA is an independent predictor of all-cause mortality; the result still remains after propensity score matching. Moreover, they found that the HUA increases the predictive power of the GRACE score and comorbidities.
The paper is interesting, even though little adds to current knowledge, but it suffers from several concerns.
The “Methods” section is challenging to “navigate” within.
i) with that high number of patients, testing the distribution of the continuous variables for the choice of statistic test is unnecessary. The statistic test simply follows the way to present the results. I suggest modifying the sentence in rows 151-154
ii) a very high percentage of patients (about 40%) enrolled were excluded from the analysis, lacking the urate determination on admission (as written by the authors in the limitation section).
Why do they not undergo the uric acid evaluation? Is it a deviation from the original protocol?
Due to the high number of patients excluded for this reason, I think that a sentence about it must be written.
iii) the cut-off point to define HUA was derived from ROC curves analysis and it differs from the values currently accepted for HUA definition (DOI:10.1016/j.jjcc.2022.04.009). It is slightly different. Why decide the authors to be confident in their ROC analysis and not accept the currently accepted cut-off point (7 mg/dl for men and 6 mg/dl for women)?
iv) the authors compared four models, but no description in the “Methods” section was provided about those models. A description of the models (though it was seen in a table of the results) is needed in the “Methods” section.
v) The propensity score is the probability of receiving one of the treatments being compared, given the measured covariates. Covariates are the variables included in the study that are not the outcome or the exposure of interest; they could be confounders or not. The propensity score is calculated by fitting a logistic regression model with treatment received or exposure to a risk factor (hyperuricemia in this case) as the dependent variable. A logistic regression model measures the change in the likelihood of a specific dependent variable given a set of independent variables (DOI: 10.1007/s11999-015-4239-4).
I think that a brief description of the calculation of the propensity score was needed
vi) about the comparison of the models is not clear: 1) what the authors compared (logistic models, Cox models, deviance, logit residuals ecc…); 2) comparing two ROC curves, which statistic test was used; please specify 3) if the Cox models were used for those comparisons (as it seems to me), why AUC, NRI, and IDI were used and not Akaike information criterion (AIC) statistically more correct (below a table for models comparisons and supporting literature).
|
Regression type |
Model comparison measure (model vs model) |
|
Logistic regression |
Deviance, LR test for deviance, AIC, BIC, AUC, NRI |
|
Cox regression |
AIC |
References
- Kleinbaum. Applied Regression Analysis and Other Multivariable Methods
- Hosmer. Applied Logistic Regression.
- Harrell. Regression Modeling Strategies.
vii) how the proportionality assumption was checked?
viii) how at multivariable analysis, collinearity was checked?
ix) how CKD was measured? Please specify the eGFR formula
x) using R for the analysis, please specify the packages used to perform the statistic tests.
“Results” section
The AUC of urates are slightly over 0.5 thus, maybe statistically, but clinically poorly significant and they can be omitted. It was better to limit that description to sensitivity and specificity deleting rows from 245 to 248
The patients were treated with PCI+DES within 48 hours from admission but half were STEMI. Analyzing a mixed cohort with different types of AMI, due to the different outcomes of one type of AMI vs another, can be crucial that the authors clarify: 1) what type of STEMI (ekg site; Tab 1); 2) time door-to-ballon of these patients; 3) systolic function of the STEMIs; 4) in the propensity score calculation, the balance of type of STEMI.
In the figure n.3, 2 ROC were compared but we read in the text only 1 C-statistic (row 252): I think that is better to specify C-statistic A (Grace) vs C-statistic B (Grace+HUA) in the text or delete it leaving only the figure. Moreover, the 2 values (in the figure) are very close with CI showing an overlap vs the concurrent value. Is really correct the p-value? I am surprised. In figure n.3 the two ROC curves appear to be very similar.
In the same way, the C-statistics of the models were specified in the text (alone without a match, making more difficult the reading). I suggest deleting the text (from row 250 to row 269) maintaining the table only and making the table and the comparisons more readable.
What NRI was used (continuous, binary)? I think that a binary NRI (with percentages of positive but also negative and global reclassification) helps to understand the real clinical significance of HUA (if logistic models are the object of the comparisons).
Round 2
Reviewer 2 Report
After the revision, the paper is improved and I have no further comments
Author Response
Dear Reviewer 2,
I appreciate your detailed and sincere responses. Thanks to your response, we improved the quality of our manuscript and learned more.
with best regards,
On behalf of the authors and with best regards.
Kwan Yong Lee, MD., PhD.
Cardiovascular Center and Cardiology Division, Seoul St. Mary’s Hospital, College of Medicine, The Catholic University of Korea, Seoul, Republic of Korea
Tel.: +82 2 2258 1139; Fax: +82 2 2258 1142, Email: cycle0210@gmail.com